# Entropic associative memory for manuscript symbols

**Rafael Morales**[1], **Noé Hernández**[2,3], **Ricardo Cruz**[2,4], **Victor D. Cruz**[3], **Luis A. Pineda**[2,3]*

**1** Universidad de Guadalajara, SUV, Guadalajara, Jalisco, Mexico, **2** Universidad Nacional Autónoma de México (UNAM), IIMAS, Mexico City, Mexico, **3** Posgrado en Ciencia e Ingeniería de la Computación, UNAM, Mexico City, Mexico, **4** Catedrático CONACyT, Mexico City, Mexico

* lpineda@unam.mx

**Data Availability Statement:** All relevant data are within the manuscript and uploaded to public repositories. Please see the following: Project repository: https://github.com/eam-experiments/EMNIST. Public links to the employed public

## Abstract

Manuscript symbols can be stored, recognized and retrieved from an entropic digital memory that is associative and distributed but yet declarative; memory retrieval is a constructive operation, memory cues to objects not contained in the memory are rejected directly without search, and memory operations can be performed through parallel computations. Manuscript symbols, both letters and numerals, are represented in Associative Memory Registers that have an associated entropy. The memory recognition operation obeys an entropy trade-off between precision and recall, and the entropy level impacts on the quality of the objects recovered through the memory retrieval operation. The present proposal is contrasted in several dimensions with neural networks models of associative memory. We discuss the operational characteristics of the entropic associative memory for retrieving objects with both complete and incomplete information, such as severe occlusions. The experiments reported in this paper add evidence on the potential of this framework for developing practical applications and computational models of natural memory.

## 1 Functional description

In this paper we present a set of experiments for registering, recognizing and recovering representations of calligraphic letters, both capital and lower case, and numerals, using the Entropic Associative Memory (EAM), with satisfactory results. Our motivation is to add evidence on the viability of the EAM model and its associated theory for developing computational models of natural memory and to show its feasibility for constructing practical applications. The formal specification of the EAM system is introduced in our previous work [1]. Here we provide an intuitive description to make the present paper self-contained. An EAM system contains a set of Associative Memory Registers (AMRs) consisting of standard tables, where columns are interpreted as the attributes or characteristics of the stored objects, the rows as their potential discrete values, and marked cells at the intersections specify the values of the attributes. Individual objects are represented as functions from attributes to values, and classes are represented through the superposition in the table of the functions representing the individuals in the class. Hence, classes are relations from attributes to values. Here we refer to the attributes in the columns and the values in the rows as the *arguments* and the *values* respectively.

resources: http://yann.lecun.com/exdb/mnist/;
https://github.com/DimaKrotov/Dense_
Associative_Memory.

**Funding:** The authors received support from the
grant PAPIIT-IN112819. The funders had no role in
study design, data collection and analysis, decision
to publish, or preparation of the manuscript.

**Competing interests:** The authors have declared
that no competing interests exist.

Each AMR of size $n \times m$ has an associated auxiliary register defined as a table of the same
dimension that is used to place the memory cue for the memory register and recognition oper-
ations, designated here as $\lambda$ and $\eta$, and also the objects extracted from the AMR by the memory
retrieval operation, designated as $\beta$. The $\lambda$-operation is defined as the logical inclusive disjunc-
tion between the value of each cell of the auxiliary register, and the value of the corresponding
cell of the AMR, for all cells, and writing up the result on the AMR. The operation can be inter-
preted diagrammatically as overlapping the content of the auxiliary register –the cue– on the
AMR proper –the memory. Fig 1 (a) shows the input of an object on an empty AMR and Fig 1
(b) illustrates the union of two functions producing a distributed representation which
includes the functions input explicitly —i.e., $\{(a_1, v_1), (a_2, v_2), (a_3, v_4), (a_4, v_7)\}$ and $\{(a_1, v_3),$
$(a_2, v_2), (a_3, v_6), (a_4, v_7)\}$— and two additional functions which are produced as collateral
effects of the $\lambda$-operation —i.e., $\{(a_1, v_1), (a_2, v_2), (a_3, v_6), (a_4, v_7)\}$ and $\{(a_1, v_3), (a_2, v_2), (a_3, v_4),$
$(a_4, v_7)\}$. The novel objects can be seen as the gain or generalization of the $\lambda$-operation. These
representations are distributed because the relation between the cells and the represented
objects are *many-to-many*; i.e., each marked cell can contribute to the representation of differ-
ent units of content –the represented objects– and each represented objects can share memory
cells with other represented objects [2].

The $\eta$-operation is defined as the logical material implication between the corresponding
cells of the auxiliary register and the AMR, the antecedent and the consequent, respectively.
This operation is interpreted as the inclusion of the cue in the AMR, and fails whenever the
auxiliary register has at least one cell *on* but its corresponding cell in the AMR is *off*. This is, if
the cue is included in the memory, the value of $\eta$ is true and false otherwise, as shown in Fig 2
(a) and 2 (b), respectively.

The $\beta$-operation, in turn, extracts the representation of an object if the $\eta$-operation is suc-
cessful for the given cue. To this effect, a value for each attribute is selected randomly out of
the set of values that are on in the corresponding column using a triangular probability distri-
bution with the cue as mode. In the present experiment the lower $v_l$ and upper $v_u$ values are
chosen such that the range $[v_l, v_u]$ is the group of values set on in the surroundings of the cue
—that is, either $l = 1$ or the value $v_{l-1}$ is off in the column, and $u = m$ or the value $v_{u+1}$ is off in
the column. In the basic case, the object retrieved is the cue exactly, as shown in Fig 3 (a); how-
ever, the retrieved object may be a previously stored object that is associated to the cue, or a
novel object constructed on the basis of the cue, as illustrated in Fig 3 (b) and 3 (c),
respectively.

AMRs have an associated entropy, which is defined as the average indeterminacy of the dis-
tributed representation that it holds. Let $\mu_i$ be the number of values assigned to the argument
$a_i$ in the relation $r$ held in an AMR; let $v_i = 1/\mu_i$ and $n$ the number of arguments in the relation's
domain. In case there are columns that have no marks, i.e., the relation is partial, we define $v_i$
$= 1$ for all $a_i$ that has no value assigned in $r$, as such a fact is fully determined. The *computa-
tional entropy* $e(r)$ –or the entropy of a relation– is defined here as:

$$e(r) = -\frac{1}{n} \sum_{i=1}^{n} \log_2(v_i).$$

The representation containing a single function is fully determined, hence its entropy is zero.

The number of functions or patterns stored in an AMR of size $n \times m$ with entropy $e$ is $(2^e)^n$
or $2^{en}$, out of the maximum AMR's capacity of $m^n$. For instance, the entropy of the abstraction
produced through the $\lambda$-operation in Fig 1 (b) is $e = 1/2$; the number of stored functions,
including the two input explicitly and the ones that are produced as side effects of the $\lambda$-opera-
tion, is $2^{(1/2)4} = 4$, out of the maximum capacity of $7^4 = 2401$. The novel functions represent

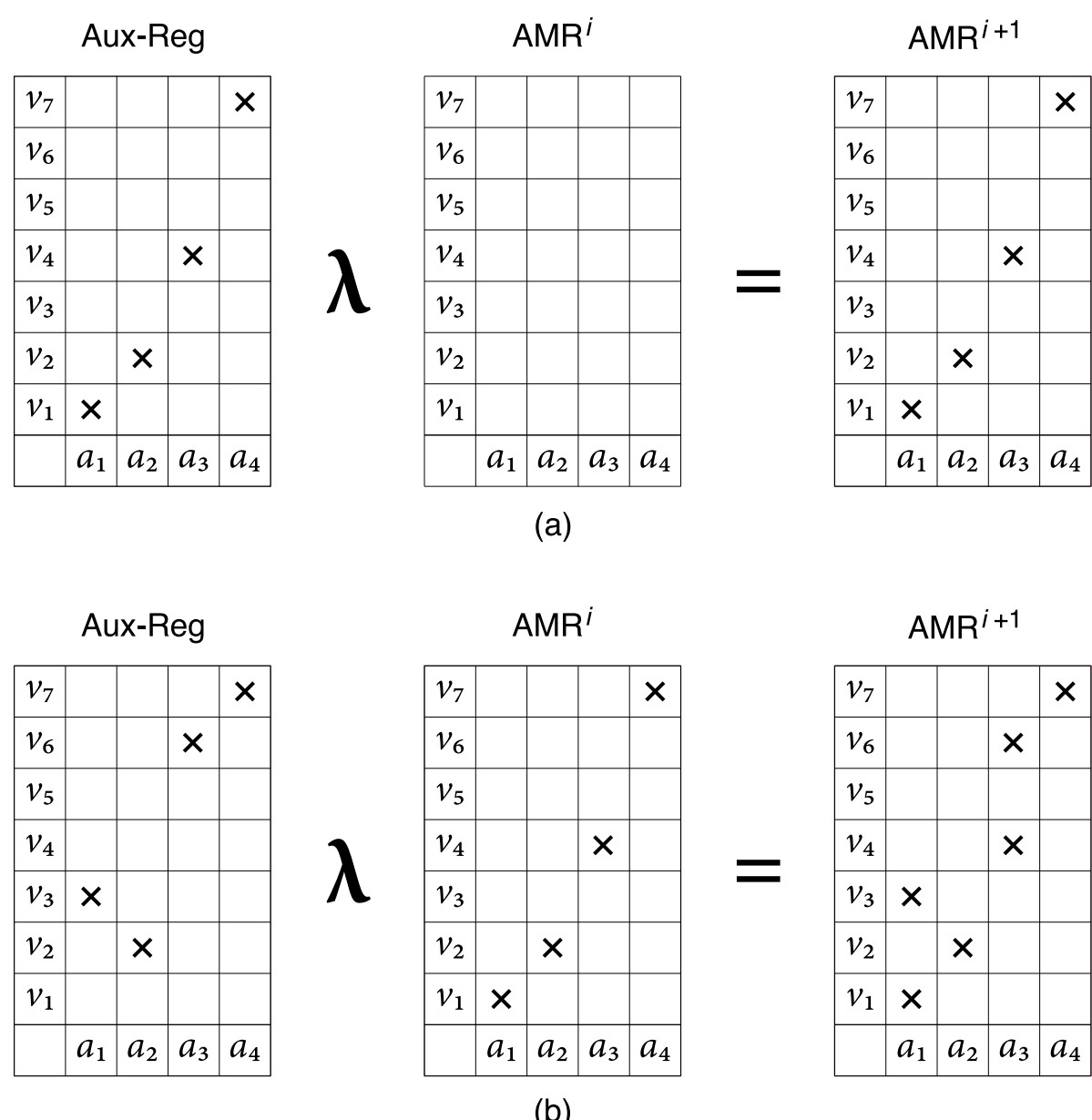

**Fig 1. Memory register operation.**

potential reconstructions that depend on the interactions between stored objects. The space of potential objects is huge and can play a role in imagination and creativity.

In summary, in the present experiments we use the original machinery of the EAM model [1] as follows: Let $r_f$ and $r_a$ be two arbitrary relations from $A$ to $V$ held in an AMR and its associated auxiliary register, respectively; and $f_a$ be a function with the same domain and codomain, held in the auxiliary register, representing the cue to a memory retrieval operation. The operations are defined as follows:

- Memory Register: $\lambda(r_f, r_a) = q$, such that $Q(a_i, v_j) = R_f(a_i, v_j) \lor R_a(a_i, v_j)$ for all $a_i \in A$ and $v_j \in V$ –i.e., $\lambda(r_f \, r_a) = r_f \cup r_a$.

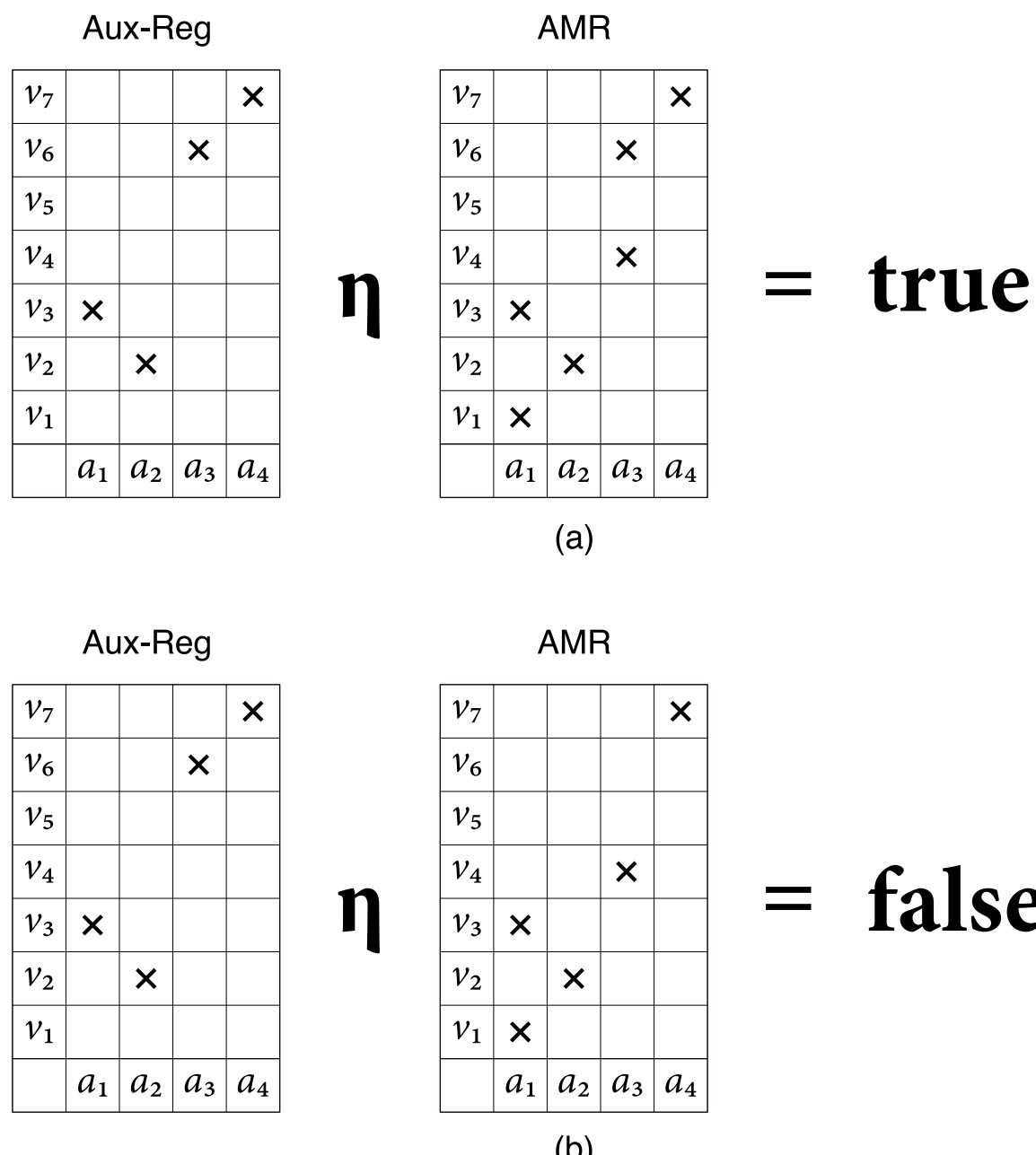

**Fig 2. Memory recognition operation.**

- Memory Recognition: $\eta(r_a, r_f)$ is true if $R_a(a_i, v_j) \rightarrow R_f(a_i, v_j)$ for all $a_i \in A$ and $v_j \in V$ (i.e., material implication), and false otherwise.

- Memory Retrieval: $\beta(f_a, r_f) = f_v$ such that, if $\eta(f_a, r_f)$ holds $f_v(a_i) = r_f(a_i)$ for all $a_i$, where such object is selected using a triangular random distribution centered on the cue $f_a$. If $\eta(f_a, r_f)$ does not hold, $\beta(f_a, r_f)$ is undefined –i.e., $f_v(a_i)$ is undefined– for all $a_i$.

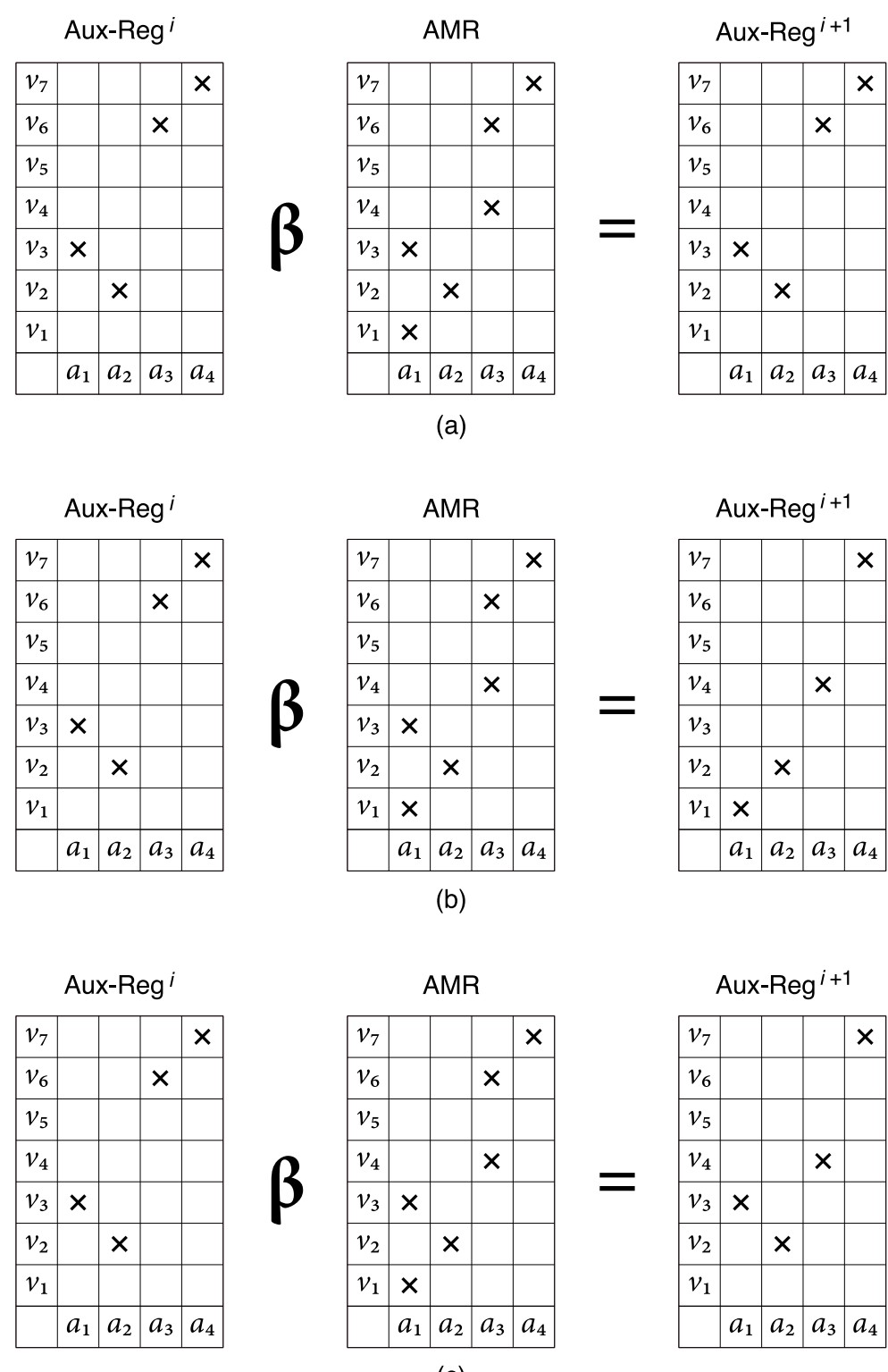

**Fig 3. Memory retrieval operation.**

## 2 Analysis and synthesis

Modality specific images are input and stored in modality specific buffers, such as standard pixels buffers for storing pictures. In the present model such concrete representations are translated into abstract amodal representations in the form of attribute-value structures or functions, which are the objects registered, recognized and retrieved in AMRs. The concrete and abstract representations of the same object stand in a *one-to-one* relation to each other; the memory regions allocating these objects are mutually exclusive and constitute *local* representations [2]. However, when such individual objects are input into AMRs, their representations are overlapped, their identity is left indeterminate and the relation between the memory units –i.e., the cells in the table– and the corresponding functions or units of content is *many-to-many*, and such representations are *distributed* [2].

The use of the present associative memory system requires an analysis module for mapping concrete into abstracts representations, for memory register and recognition, and a synthesis module for performing the inverse mapping. The bidirectional mapping is performed through an autoencoder [3, 4] including an encoder and a decoder [5]. These two components are modular –independent– and implement the analysis and synthesis modules, respectively, as illustrated in Fig 4.

The encoder in the experiments presented below in Section 3 is implemented with a VGG5-like neural network with ten convolutional layers [6], and the decoder with a transposed convolutional neural network with four layers. The classifier is a fully connected neural network (FCNN) with two layers, mapping sets of features output by the encoder into their corresponding $c$ classes, and its purpose is to moderate the autencoder's training process, so that the abstract representations are satisfactory both for classification and decoding purposes. The architectures for the neural networks were chosen so that they were simple but functional.

The input to the encoder consists of $p$ features, that correspond to the information in the input buffer, as illustrated in Fig 4. The encoder maps such concrete representation into $n$ outputs, which constitutes the input to the AMRs directly –i.e., a function of $n$ arguments with their corresponding values. This set of features is also the input to the classifier, which associates one of $c$ categories to its input, and the decoder, that computes an approximation of the inverse function of the encoder, and renders $p$ features, regenerating the concrete image. The encoder, the decoder and the classifier are trained simultaneously in a supervised manner by standard back-propagation, and the latter is removed once the autoencoder has been trained. Autoencoders were originally proposed to reduce the dimensionality of the data [3, 4], and the present use constitutes a novel application of such technology.

The $n$ outputs of the encoder are floating point values which are converted into integers in the range $[1, m]$ by rounding the results of a linear transformation, producing the discrete value of the corresponding argument. The resulting discrete function is input into the corresponding AMR in the memory register operation or is used as the cue to the memory recognition or retrieval operation. There are $c$ associative memories, one per class in the dataset. If the cue is accepted the corresponding object is retrieved as well as its class; otherwise a conventional rejection code is returned with the class *unknown*. If the cue is accepted by more than one AMR the class with the smallest entropy is selected. This operation is performed by the *entropy-based filter*.

## 3 A visual memory for hand written manuscript symbols

The associative memory system was tested in previous work [1] through the construction of a visual memory for storing and retrieving distributed representations of hand written digits from "0" to "9". The system was built and tested using the MNIST data-set available at

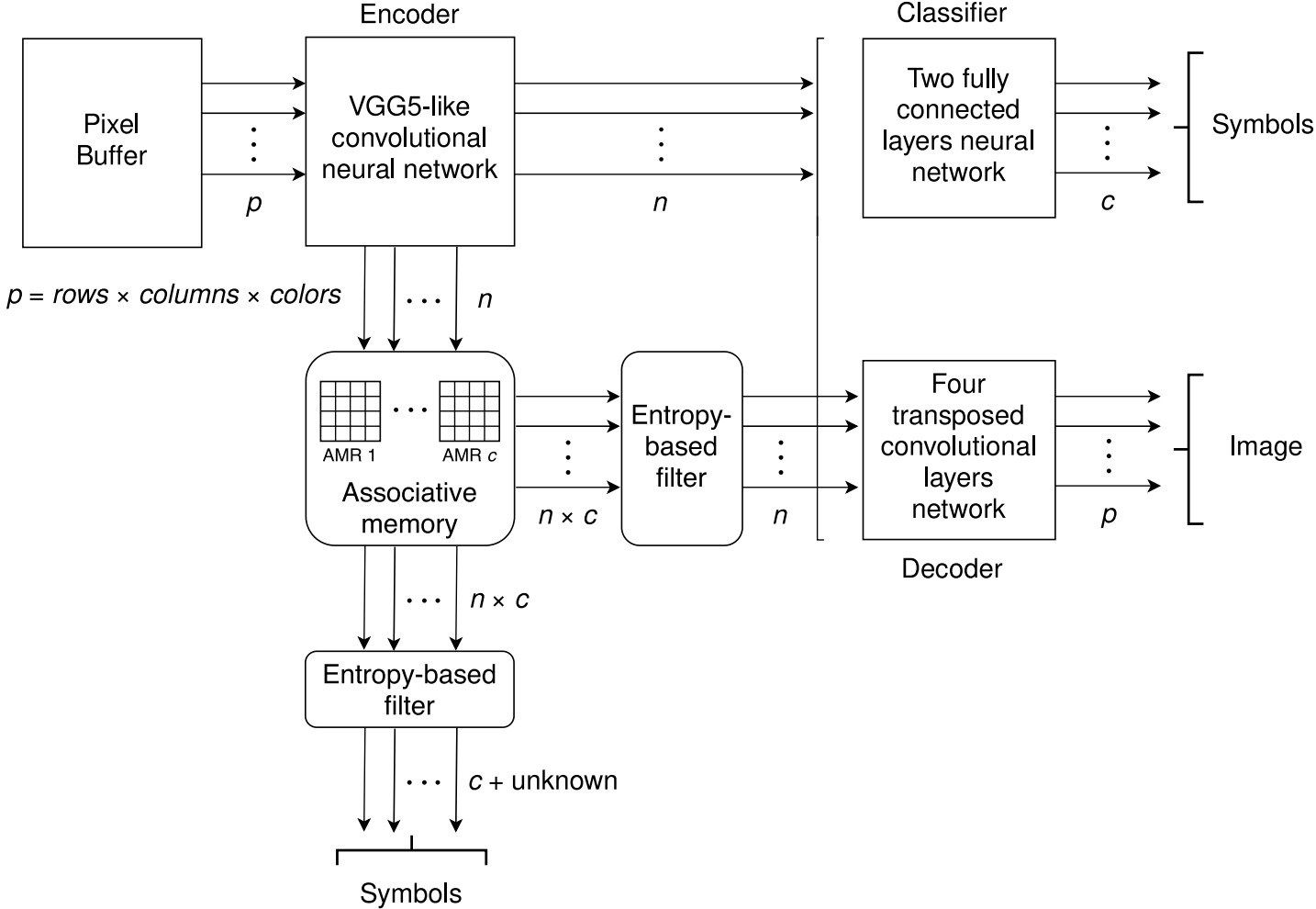

**Fig 4. Training the analysis and synthesis modules.**

http://yann.lecun.com/exdb/mnist/. In the present work we add evidence of the potential of the framework for the construction of practical applications. We use now the EMNIST [7] database including manuscript capital and lower case letters, and also the ten digits, increasing the number of classes in relation to MNIST from 10 to 62; however, capital and lower case letters with very similar visual shapes are further merged, rendering 47 classes. Here we define the alphabet EMNIST-47, as shown in Fig 5. As can be seen, there are eleven lower case letters that can be clearly distinguished from their upper case counterparts –i.e., classes 36 to 46– and have independent entries. For the experiment 1 described in Section 3.1 we used the EMNIST Balanced segment including 2, 800 instances of each class.

In previous work we also showed that an Associative Memory Register can hold the representation of more than one class and yet the system has a satisfactory performance. Here we capitalize such functionality and also model a memory in which capital and lower case letters of the same type that have very different visual shapes are held in the same memory register. As a result, classes 36 to 46 of EMNIST-47 are dropped, and the total number of classes is reduced to 36. This second alphabet is referred to as EMNIST-36, and it is shown in Fig 6.

| Class number | Class name | Class instances | Class number | Class name | Class instances | Class number | Class name | Class instances |
|---|---|---|---|---|---|---|---|---|
| 0 | 0 | | 16 | G | | 32 | W | |
| 1 | 1 | | 17 | H | | 33 | X | |
| 2 | 2 | | 18 | I | | 34 | Y | |
| 3 | 3 | | 19 | J | | 35 | Z | |
| 4 | 4 | | 20 | K | | 36 | a | |
| 5 | 5 | | 21 | L | | 37 | b | |
| 6 | 6 | | 22 | M | | 38 | d | |
| 7 | 7 | | 23 | N | | 39 | e | |
| 8 | 8 | | 24 | O | | 40 | f | |
| 9 | 9 | | 25 | P | | 41 | g | |
| 10 | A | | 26 | Q | | 42 | h | |
| 11 | B | | 27 | R | | 43 | n | |
| 12 | C | | 28 | S | | 44 | q | |
| 13 | D | | 29 | T | | 45 | r | |
| 14 | E | | 30 | U | | 46 | t | |
| 15 | F | | 31 | V | | | | |

**Fig 5. EMNIST-47 alphabet.**

The corpus was partitioned for the experiments in both settings in three disjoint sets. The partitions were rotated through a standard 10-fold cross-validation procedure. The partitions names and the assigned amount of data are as follows:

- Training Corpus (*TrainCorpus*): For training the analysis and synthesis modules (57%).

- Remembered Corpus (*RemCorpus*): For filling in the Associative Memory Registers (33%).

- Test Corpus (*TestCorpus*): For testing (10%).

The data allocated to each partition reflects a trade-off between learning and test data, so 90% of the corpus is used for the former and 10% for the latter, according to standard machine learning practices. The balance between training and remembered data considers that a large enough amount of corpus is needed for training the deep neural networks modeling perception and action, but it is also required a large enough amount of data to test the AMRs with different remembering conditions and entropy levels. We also made preliminary experiments with small variations to these amounts, and the present choice constitutes a satisfactory compromise.

The functional neural network architectures and the optimal value for the number of inputs to the AMRs –i.e., the parameter $n$– was determined using the training corpus through preliminary experiments. The integer powers of 2 were explored and $n$ was set to 64 for all the experiments. Next the training corpus was used for training the neural networks –the encoder, the decoder, and the classifier, assembled into a single neural network with one input and two output channels– for all the tend folds. Then the encoder was used to process the remembered

| Class number | Class name | Number of instances | Class instances | Class number | Class name | Number of instances | Class instances |
|---|---|---|---|---|---|---|---|
| 0 | 0 | 2,800 | | 18 | I | 2,800 | |
| 1 | 1 | 2,800 | | 19 | J | 2,800 | |
| 2 | 2 | 2,800 | | 20 | K | 2,800 | |
| 3 | 3 | 2,800 | | 21 | L | 2,800 | |
| 4 | 4 | 2,800 | | 22 | M | 2,800 | |
| 5 | 5 | 2,800 | | 23 | N | 5,600 | |
| 6 | 6 | 2,800 | | 24 | O | 2,800 | |
| 7 | 7 | 2,800 | | 25 | P | 2,800 | |
| 8 | 8 | 2,800 | | 26 | Q | 5,600 | |
| 9 | 9 | 2,800 | | 27 | R | 5,600 | |
| 10 | A | 5,600 | | 28 | S | 2,800 | |
| 11 | B | 5,600 | | 29 | T | 5,600 | |
| 12 | C | 2,800 | | 30 | U | 2,800 | |
| 13 | D | 5,600 | | 31 | V | 2,800 | |
| 14 | E | 5,600 | | 32 | W | 2,800 | |
| 15 | F | 5,600 | | 33 | X | 2,800 | |
| 16 | G | 5,600 | | 34 | Y | 2,800 | |
| 17 | H | 5,600 | | 35 | Z | 2,800 | |

**Fig 6. EMNIST-36 alphabet.**

and test corpora to produce the corresponding sets of abstract representations. These were in turn input to the corresponding AMRs to perform the experiments.

Developing on our previous work, the entropic associative memory for manuscript symbols was tested with four experiments as follows:

1. Experiment 1: Define an associative memory system including an AMR for holding the distributed representation of each one of the forty seven manuscript symbols of EMNIST-47. Determine the recognition precision and recall of the individual AMRs, and of the overall system, for AMRs of size $n \times 2^m$. The experiment was performed for $0 <= m <= 9$. Identify the parameter $m$ of the AMRs with satisfactory performance. Finally, determine the precision and recall of the memory recognition when the AMRs contain different amounts of remembered instances and, consequently, different levels of entropy.

2. Experiment 2: Determine the value of $m$ of AMRs holding the distributed representation of capital and lower case letters of EMNIST-36. Determine the recognition precision and recall of the individual AMRs and of the overall system, for $0 <= m <= 9$ as before. Determine the precision and recall of the memory recognition when the AMRs contain different levels of entropy, as in experiment 1.

3. Experiment 3: Retrieve objects out of a cue for different levels of entropy and generate their corresponding images –with the best AMRs found in experiment 1. Assess the similarity between the cue and the recovered object at different levels of entropy.

4. Experiment 4: Retrieve letters and digits of significantly occluded objects with the same AMR size used in Experiment 3. Assess the precision and recall of the memory retrieval operation and the quality of the generated images.

The source code for replicating the experiments, including the detailed results and the specifications of the hardware used, are available in Github at https://github.com/eam-experiments/EMNIST.

### 3.1 Experiment 1

Compute the characteristics of AMR of size $64 \times 2^m$ for $0 \leq m \leq 9$:

1. Register the totality of *RemCorpus* in their corresponding register through the *Memory_Register* operation;

2. Test the recognition performance of all the instances of the test corpus through the *Memory_Recognize* operation;

3. Compute the average precision, recall and entropy of individual memories.

4. For each instance of the test corpus recover a unique object by the *Memory_Retrieve* operation or reject the cue; compute the average precision and recall of the integrated system when this choice has been made.

5. Select the parameter *m* with the best trade-off between precision and recall. Determine the performance of the system with such memory size $n \times 2^m$, for different amounts of the *RemCorpus* and entropy levels.

The average precision, recall, and entropy of the AMRs, across the ten-fold cross-validation experiment is shown in Fig 7 (a). Precision and recall are computed per AMR in the standard way as follows: *Precision* = *TP*/(*TP* + *FP*); *Recall* = *TP*/(*TP* + *FN*), where *TP*, *FP* and *FN* stand for true positives, false positives and false negatives, respectively. As can be seen, precision is very low for small values of *m* but grows with the number of rows, i.e., $2^m$, but recall is very high for low values of *m*, as the information is confused when the number of rows is very low, and most instances are accepted. However, when the value of *m* is increased, the grid is made finer, true instances are missed, and the recall is lowered. The optimal value of *m* is a trade-off between the precision and recall graphs. Fig 7 (a) shows that there is a good compromise at *m* = 6 and *m* = 7, i.e., for 64 and 128 rows. The figure also shows the entropy at the bottom bar. As can be seen its value is increased almost linearly with the AMRs size, starting from 0 for 1 row, and is maximal for 512 rows.

Fig 7 (b) shows the precision and the recall as a function of the number of rows of the system as a whole. In this case if an instance is rejected by all AMRs, it counts as a false negative for the memory system, and lowers the total recall. In case the instance is accepted by at least one AMR, the one with the lowest entropy is chosen; if it is of the wrong class, it is a false negative for the right class and a false positive for the accepting class, and increases by one the count of both the false positives and false negatives of the system as a whole. Fig 7 (b) shows that precision and recall have a very similar pattern, as both grow from a very low to a higher value according to the increase of the number of rows. However, when the grid is too fine, true instances are reject and recall starts to decrease as before. Fig 7 (a) and 7 (b) are coherent, and both show that there is a good compromise at 64 and 128 rows. Fig 7 (c) shows the average

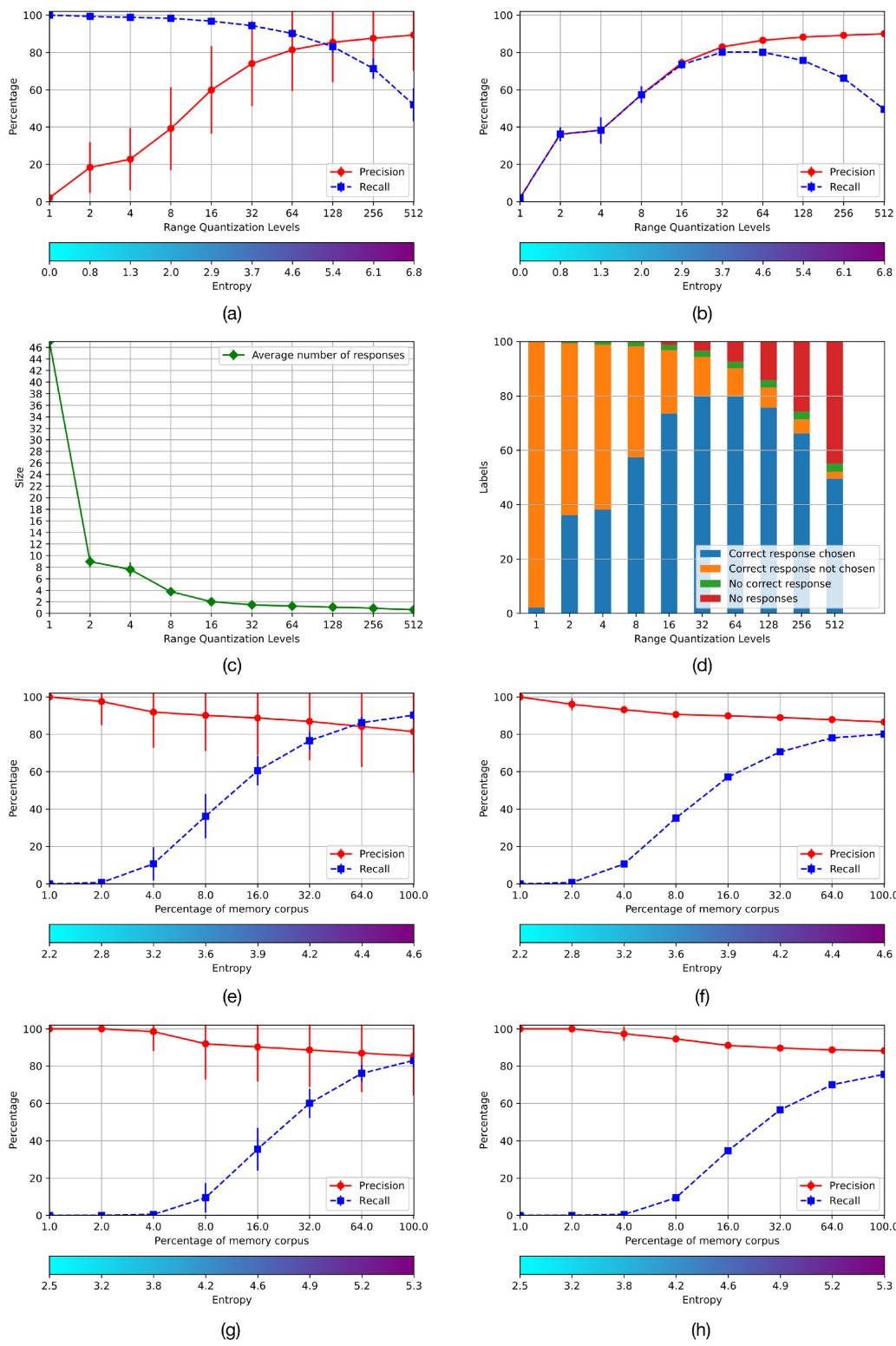

**Fig 7. Results of Experiment 1 using EMNIST-47.**

number of accepting AMRs for each instance per AMR size. As can be seen, this number goes from 47 for AMRs with one row to 1 for AMRs with 32 or more rows. This effect is further illustrated in Fig 7 (d).

The overall purpose of experiment 1 was to investigate the performance of AMRs with satisfactory operational characteristics in relation to its entropy or information content. These are the AMRs with sizes $64 \times 64$ and $64 \times 128$. Fig 7 (e) and 7 (g) show the respective average performance of these AMRs when they are filled up with varying proportions of the *RemCorpus* – 1%, 2%, 4%, 8%, 16%, 32%, 64% and 100%– using the best AMRs size; and Fig 7 (f) and 7 (h) show the performance of the system as whole for the corresponding register sizes, as before.

These latter figures show another aspect of the entropy trade-off. If the number of remembered instances is very low, the entropy is also very low –in the limiting case, if there is only one instance, the entropy is zero. In this condition there are very few objects stored in the AMRs, and if the cue to the memory recognition operation is accepted, precision is very high; however, even cues that are very similar to the stored objects will be rejected; hence recall is very low. Precision decreases slightly with the increase of remembered information and the consequent entropy increase, but recall grows quite rapidly, until a satisfactory compromise between precision and recall is reached. However, if the entropy is increased even further the precision starts to lower but the recall continues to grow. In this case the AMRs are saturated, most instances are accepted and recall is very high but precision lowers significantly.

Fig 7 (e) shows that the best trade-off between precision and recall for AMRs of size $64 \times 64$ occurs when the percentage of the remembered corpus included in the memory is 64% where the corresponding graphs intersect. The graphs vary slightly for AMRs of size $64 \times 128$ where they do not intersect, and the best performance occurs when the totality of the remembered corpus is used, as shown in Fig 7 (g). The performance of the system as a whole, shown respectively in Fig 7 (f) and 7 (h), shows that the precision remains high when the totality of the remembered corpus is used, but the recall is lower and the graphs do not intersect. To asses the performance of the system it also has to be considered the cost of the memory resource which is doubled for the largest register.

### 3.2 Experiment 2

This experiment shows that an AMR can hold the distributed representation of objects of different classes adding evidence on previous work, such as the representations of different digits –e.g., 0 and 1– with similar levels of precision and recall but a small increment of the entropy, as was shown in our previous work. In this case, we use EMNIST-36 instead of EMNIST-47 such that eleven different shaped capital and lower case letters are collapsed in the same class. The procedure is analogous to experiment 1. The results are shown in the corresponding graphs in Fig 8. The performance of the two settings are analogous, with the only difference that the entropy of the AMRs holding capital and lower case letters is slightly larger than the entropy of the corresponding AMR holding only one class.

Experiments 1 and 2 show that the performance of the systems is mostly similar for AMRs of sizes $64 \times 64$ and $64 \times 128$; they also show that the system's performance is similar for the alphabets EMNIST-47 and EMNIST-36. The latter is more economical –as it abstracts over lower case and capital letters and uses only 36 AMRs– hence we use AMRs of size $64 \times 64$ with the EMNIST-36 for investigating further the quality of the memory system in experiments 3 and 4.

### 3.3 Experiment 3

This experiment assesses the performance of the memory retrieval operation, using the configuration and entropy levels in Fig 8 (e) and 8 (f). This operation is constructive in opposition to

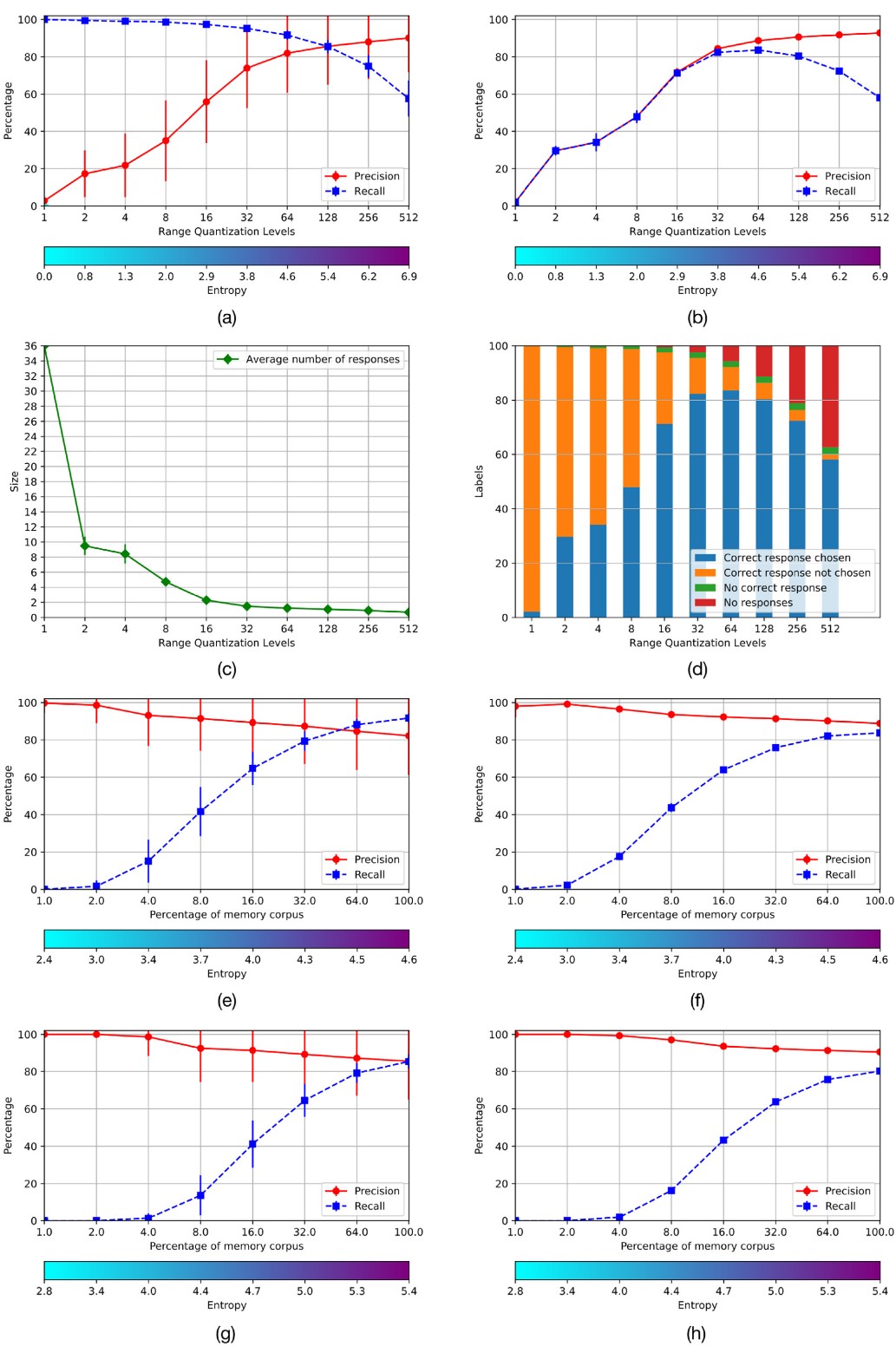

**Fig 8. Results of Experiment 2 using EMNIST-36.**

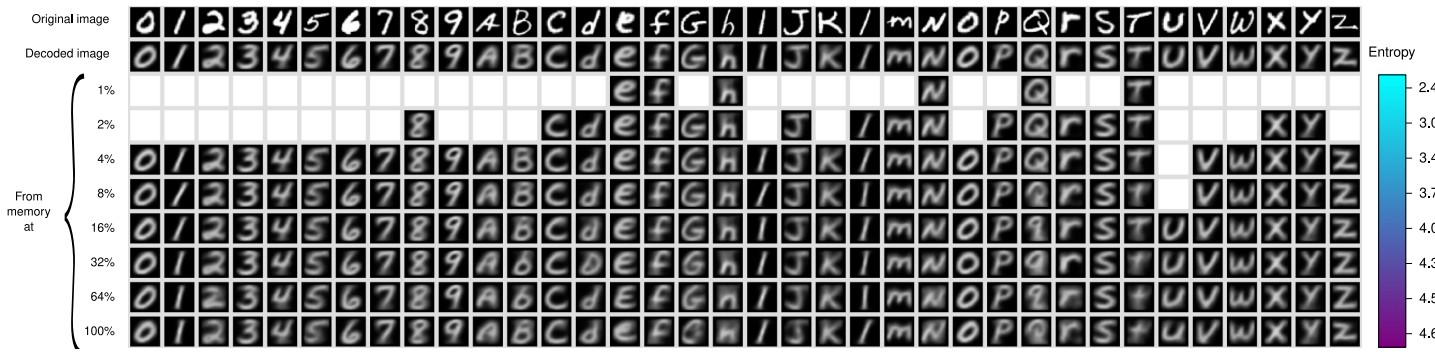

**Fig 9. High quality cues.** EMNIST-36 Symbols recovered with high quality cues as a function of the entropy.

photography memories, and all recovered objects are produced by the $\beta$-operation as explained above in Section 1 and illustrated in Fig 3. We study three conditions for memory retrieval, which are illustrated in the tables of Figs 9–11. There is a column for every type of symbol of the EMNIST-36 alphabet. The different rows show the symbol that is recovered from the memory using the same cue at the different amounts of corpus and levels of entropy. White cells indicate that the memory cue was rejected at the corresponding entropy level. The top row shows the cue to the memory retrieval operation. The second row shows the symbol

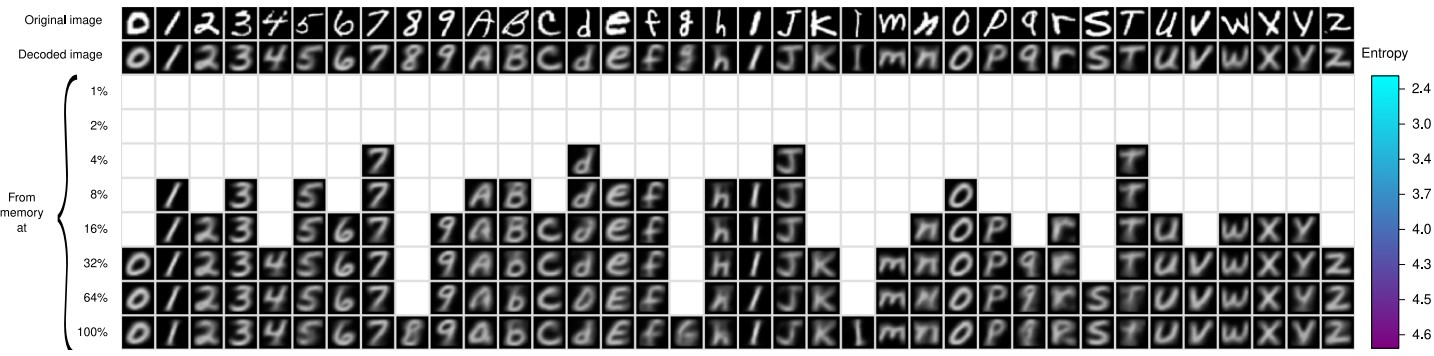

**Fig 10. Moderate quality cues.** EMNIST-36 Symbols recovered with moderate quality cues as a function of the entropy.

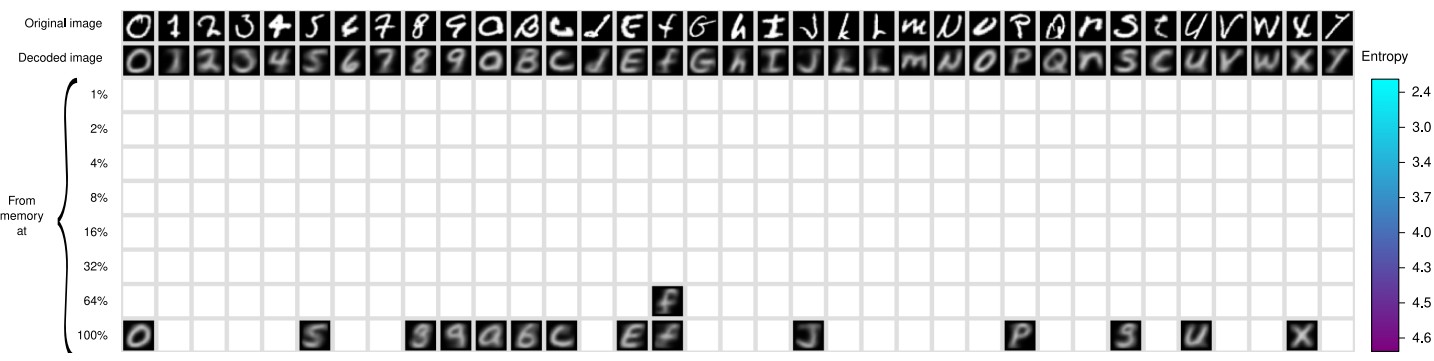

**Fig 11. Poor quality cues.** EMNIST-36 Symbols recovered with very poor cues as a function of the entropy.

that is produced by the autoencoder feeding the output of the encoder directly into the decoder, i.e., bypassing the memory. As the decoder ideally computes the inverse function that is computed by the encoder, the symbols in this row should be exact copies of corresponding cues in the top row. However, the decoder computes only an approximation and the symbols are slightly different. The encoder and decoder produced always an object, which is the most proximate to the cue, as neural networks never reject an object, despite that the quality of the cue may be very poor; hence the rendered object may be of a wrong class. The 3rd to the 10th row show the symbols recovered at the eight different levels of entropy of the *RemCorpus*. We study three conditions of the memory retrieval operation, as follows:

1. The cue to the memory retrieval operation is accepted at the lowest possible level of entropy. In this condition it is expected that the symbol will be recovered at all higher levels of entropy too, which it is in fact the case as shown in Fig 9. The table shows that the recovered object is quite similar to the cue at most entropy levels, although its quality diminishes slightly at higher levels. This condition illustrates cues that are recovered with high precision, but there are very few objects stored in the memory and recall is very low. Intuitively, the contribution of the cue to the construction of the retrieved object is very high, hence the cues and the corresponding retrieved objects are very similar and have "good quality".

2. The cue is accepted at a moderate level of entropy –at about 4% of the remembered corpus or at an entropy level between 3 and 4– but is rejected at very low levels, as illustrated in Fig 10. In this condition both the cue and the memory content contribute to the shape of the retrieved object depending of the entropy level: the higher the entropy the less the impact of the cue on the shape of constructed object and vice versa. However, as there are more objects stored in the memory, the entropy and the recall are higher, and more cues are likely to be recovered.

3. The cue is accepted at the highest entropy levels or rejected, as illustrated in Fig 11. In this condition the cues are far from representative instances of the class. If accepted, the content of the memory may contribute much more to the construction of the retrieved object than the cue itself; the reconstruction may be very noisy; the quality of the recovered objects may be poor or very poor; and the cue and the retrieved object may be quite different. Precision is very low and the retrieved object may be of the wrong class.

### 3.4 Experiment 4

In this experiment, we investigated the memory retrieval operation using severely occluded cues. In this condition, cues should be rejected at most entropy levels, as they are very different of the corresponding stored objects. This is reflected in the recall of the $\eta$-operation, and consequently of the $\beta$-operation, which is very low. However, there may be situations in which recovering tentative objects may be useful if they can be further processed in relation to contextual information, and the right object may still be recovered, as is common in visual interpretation. Despite severe occlusions, the visible part of the cue has some amount of structure that is reflected in its abstract representation, and recognition failure may be due to the rejection of a small number of features, as shown in our previous work [1]. The definition of the $\eta$-operation states that all 64 features of the abstract representation of the cue must be included in the AMR of the corresponding class, which is a very strong condition. A means to increase the recall is to relax the test and allow that memory retrieval is successful if a small number of features may fail.

We tested the memory retrieval operation when the bottom-half of the manuscript symbols were occluded, and also when the symbols were occluded by horizontal bars covering more

than half of the total area in which the symbol is placed. The results are shown in Figs 12 and 13, respectively. Figures (a) at the top in both occlusion conditions show the objects recovered with no tolerance at all levels of entropy, and figures (b), (c) and (d) show the relaxation of 1, 2 and 3 out of the 64 total features, respectively. As expected, the recall is increased but the precision is lowered according to the amount of relaxation. As the recall is very low in all these conditions only a small amount of symbols are recovered. In the none-relaxation condition most cells are blanks at low entropy levels and there are columns with only blank cells. The relaxation of one feature shows that objects are recovered at lower levels of entropy, and that the blanks are filled-up from bottom to top; this tendency is continued with the relaxation of two and three features as shown in (c) and (d) in both figures, respectively.

## 4 Experimental setting

The programming for all the experiments was carried out in Python 3.8 on the Anaconda distribution. The neural networks were implemented with TensorFlow GPU 2.4.1, and most of the graphs were produced using Matplotlib. The experiments were run on an Alienware Aurora R5 with an Intel Core i7-6700 Processor, 64 GBytes of RAM and an NVIDIA GeForce GTX 1080 graphics card.

## 5 EAM versus neural networks models of associative memory

Figs 1 to 3 illustrate that EAM resembles natural memories [8, 9] in a number of putative properties, and diverges from the proposals developed within the neural networks paradigm [10–16], that were consolidated with Hopfield's model [17], and subsequent work [18–23]. The basic difference is that we conceive a memory as a functional system using a declarative representational format in which recollections are registered, recognized and recovered on the basis of a cue, which corresponds to the intuitive notion of "remembering", in opposition to neural networks models that conceive memory as a dynamical system in which input patterns are used to "train" a neural network in a "learning stage"; such that the stored patterns can be recovered out of complete or incomplete input patterns in a "use" or "test stage". Although all neural networks models use numerical matrix representations, they can be divided intuitively into those that focus on the memory as a dynamical system, as Hopfield's [17] and Kosko's [18] original models; and those that focus on the algebraic properties of the matrices representing such networks, such as morphological models [19, 20, 24] and related work [25]; and those inspired on natural neurons directly, such as dendritic associative memories [26]. There is also work modeling human associative memory at the hardware level in situations of conditioning, emotion and fatigue [27, 28]. The difference between the present proposal and the neural networks models shows up in several specific characteristics of the systems, as elaborated below.

### 5.1 Representational format

AMRs in the EAM model hold distributed representations that are produced out of the logical inclusive disjunction of discrete finite functions representing individual objects in the table format, and conform directly to the criterion stated by Hinton [2] for distributed representations, i.e., that units of memory stand in a *many-to-many* relation to the units of content, as opposed to local representations where such relation is *one-to-one*. This is, it is possible to tell explicitly which units of memory contribute to the representation of an individual object, and which memory cells are shared by a number of units of content or represented objects. The representations stored in the AMRs are manipulated directly by the memory register, memory recognition and memory retrieval operations, and the representation is declarative. This contrasts with memories developed in the neural networks paradigm, in which the memories are

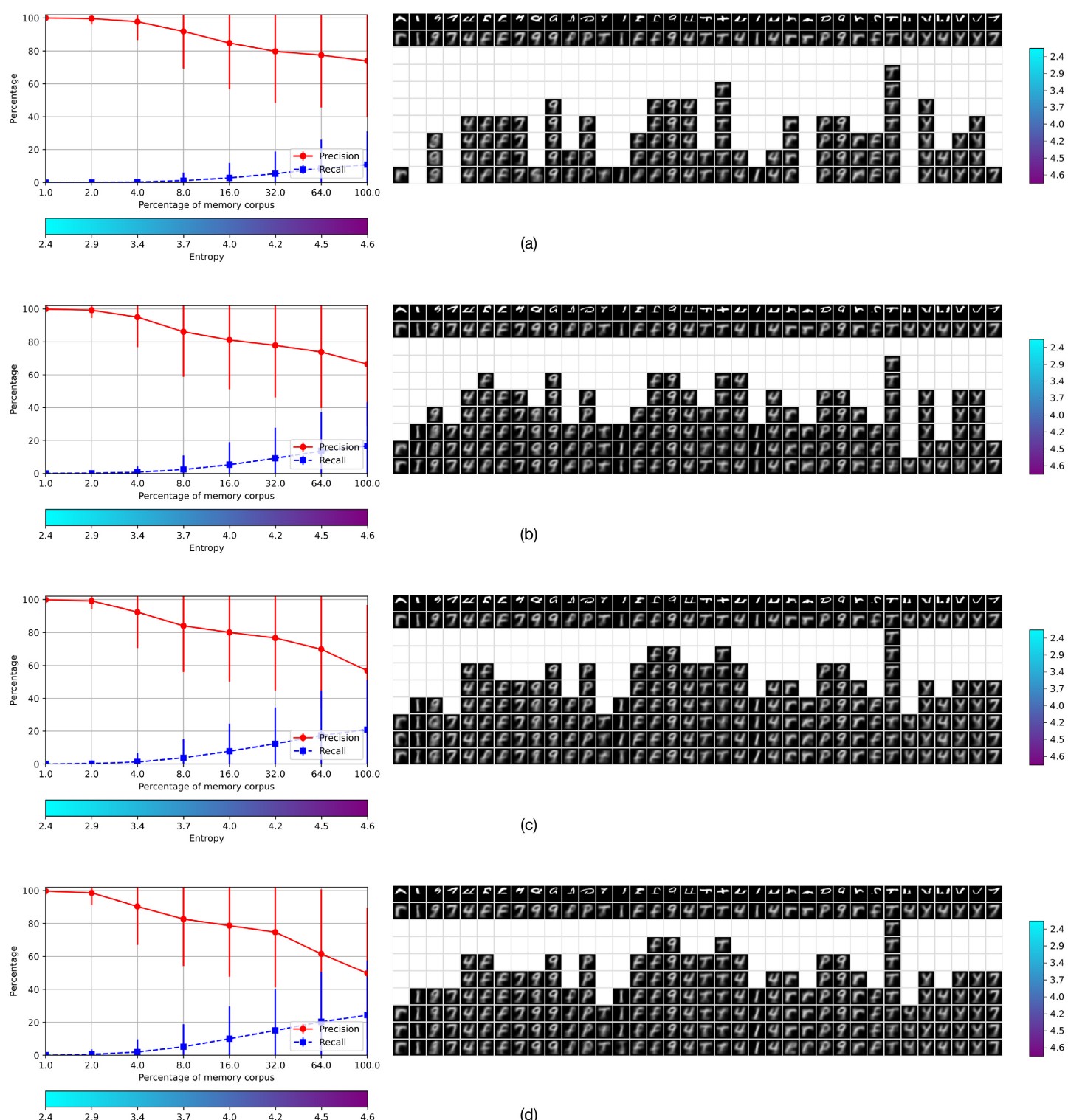

**Fig 12. Memory retrieval with occlusions of 50%.** Results of using cues occluded 50% at the bottom for memory retrieval, with relaxations of none, 1, 2 and 3 features out of 64.

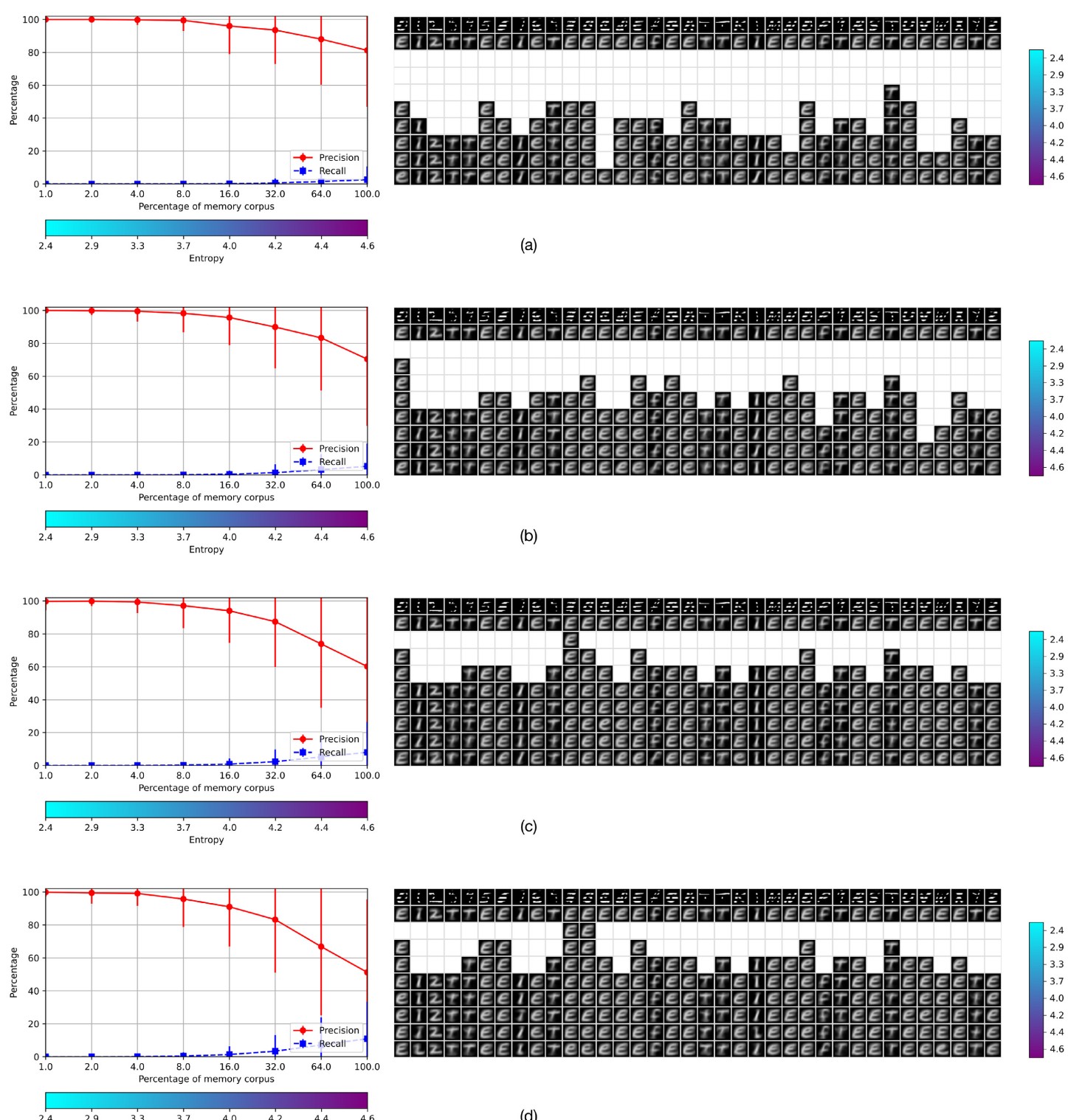

**Fig 13. Memory retrieval using cues occluded with horizontal bars.** Results with relaxations of none, 1, 2 and 3 features out of 64.

numerical matrices of weights produced through standard matrix operations; and such representations are better thought of as sub-symbolic or non-declarative.

## 5.2 Productivity

A salient property of distributed representations is that they "contain" not only the objects used in the training process, but also additional similar objects that are produced as a side-effect of the operations through which the representation is created [2]. This generalization property is essential to neural networks devices, such as classifiers trained with standard back-propagation, that are constructed with a training corpus that should be different from the testing corpus, and can classify unseen inputs in the test stage. Associative memories within the neural networks paradigm are trained through a forward process, such that each input pattern should correspond ideally to a minimum of the energy function associated to the network, and can be retrieved explicitly. Although there may by additional local minima, these are considered errors due to side-effects of the training, that should be avoided. The EAM model, on its part, do produce a potential number of novel patterns due to the productivity of the λ-operation, which define the space of potential reconstructions; hence, has the generalization property, but in a manner that is coherent with the retrieval of explicitly stored patterns.

## 5.3 Constructive recovery

Associative memories developed within the neural networks paradigm need to be provided with the full set of patterns that are codified in the weight matrix, and the retrieve operation, either on the basis or complete or incomplete inputs, produces the stored patterns exactly; for this reason, such memories are reproductive or "photographic", alike to standard RAM memories, and can be better considered pattern matching machines. This contrasts with EAM, where the objects retrieved out of a complete or a partial cue through the $\beta$-operation are constructions always. A recovered object may be identical but also similar to the cue, and even an "imaged object", resembling better the intuitions reported by Bartlett in his seminal work, and also the common intuition about remembering objects in natural memory.

## 5.4 Kinds of associations

Neural networks models of associative memory are classified as *auto-associative* and *hetero-associative*. The former are systems in which the cue is the pattern to be recovered itself, either in full or a segment of it, and possibly with noise. Hence, the images codified and stored in the memory can be recovered with partial information, e.g., when the image is partially occluded or noisy. Hopfield's [17] original model is the paradigmatic case of auto-associative memories. Hetero-associative memories store a relation between two patterns, e.g., the pair $(a, b)$, that represent associated or similar objects, possibly of different classes. In these models, $b$ can be recovered using the pattern $a$ or a partial or noisy version of it and vice versa, and the memory is said to be "bidirectional". A paradigmatic case of this model is the so-call BAM model [18]. These models require that all the associations are included in the training set; hence, associations cannot be established dynamically on the basis of the content of the memory at an arbitrary state and a novel input. In contrast, the distinction between auto and hetero associative is not basic in the EAM model. In the case study presented in Section 3 the auto-associative property is illustrated with the recovery of calligraphic letters and numerals out of complete and severely occluded cues; but, in addition, the bidirectional hetero-associative property is illustrated with the recovery of a capital letter using a lower-case letter as the cue and vice versa, which are stochastic processes. In EAM all patterns are input independently and the associations are established dynamically; this is due to the representational format, the

definition of the λ-operation, and the stochastic aspect of the *β*-operation; unlike hetero-asso-ciative models, in which associations are established explicitly.

## 5.5 Memory rejection

The EAM model defines an explicit reject operation through the logical material implication without search, implementing the true or strong negation. This property is not addressed explicitly in neural networks models of associative memory. Such models are oriented to recover the pattern that is most similar to the input. For instance, Hopfield's and Kosko's mod-els use a recurrent search process that converges to the pattern in the memory that is the most similar to the input pattern. However, if the input is partial the system may converge to the wrong pattern, and the object recovered would be a false positive. It is also possible that the process does not converge in a reasonable amount of time, and the process needs to be inter-rupted. In this situation it is possible to return the current hypothesis of the search process, which may be a false positive; an alternative may be to state that the pattern was not found, implementing a form of negation as failure [24]. These problems are avoided in systems that do not search and provide the answer through an analytical process that always terminate. Such is the case of morphological hetero-associative memories [19, 20, 22] and related work [25] which are also bidirectional. However, the full set of associations must be given in advance to train the network, and such systems do not contemplate that an input may not be included in the memory; hence, whenever this is the case, provide the most similar object, which is a false positive by necessity. More generally, the assumption that the input must converge to a known object presupposes that all of the patterns in the domain of interest are included in the training set, or that the information about the world is complete, so to approximate to the most similar object is valid. This conforms to the so-called Closed-World Assumption (CWA) in its most general form.

The EAM model, on its part, do have an explicit reject capability and implements the true or strong negation, and assumes incomplete information. This is an essential property of a nat-ural memory, because its purpose is precisely to learn or acquire novel information. Natural memory is also oriented to the interaction with the environment, which makes untenable the adoption of the CWA, and also that the memory should be trained in full whenever a new bit of information is acquired.

## 5.6 Parallelism

The three memory operations of EAM can be computed through parallel computations between corresponding cells of the AMR and its associated auxiliary register in two or three computing steps, i.e., the operation proper and updating the AMR and its auxiliary register, if the appropriate hardware is provided. Furthermore, a cue may be used to select a large number of AMRs through a BUS simultaneously. Parallel computing in neural networks models, on its part, is achieved through the parallelization of matrix operations, as any standard matrix paral-lel computing, with the corresponding demands of RAM and time.

## 5.7 Energy function

Neural networks models of associative memory have an associated energy function which determines the search path from the high to the low-energy nodes, that correspond to incom-plete cues memories and stored patterns, respectively. The definition of such function is essen-tial for models in the dynamical system's paradigm. The EAM model, on its part, does not require of such notion, because it is not construed as a dynamical system.

## 5.8 Entropy

The EAM model defines the computational entropy, which is a measure of the indeterminacy of the representation contained in an AMR. Operational AMRs obey an entropy trade-off such that there is an interval of entropy values, where the entropy is not too low nor too high, where the $\eta$ and $\beta$-operations have a good compromise between precision and recall in relation to a cue, as is shown in the case study in Section 3. The neural networks models hold fully determined patterns, do not involve a notion of indeterminacy, and do not use an operational notion of entropy.

## 5.9 Capacity

The storage capacity of Hopfield's memories is about 0.15 patterns in relation to the number of nodes. This has been extended significantly with approaches that use a more general energy function [19–21, 23, 24, 29] and the promise of these systems is that they have a very large storing capacity using moderate physical memory resources. This is also the case in the present proposal. For instance, the AMRs of size $64 \times 64$ –i.e., 4096 bits– can hold up to $64^{64}$ patterns. This capacity is modulated by the amount of information stored in AMRs, which should be not too low and not to high, obeying the entropy trade-off, that determines their operational capacity. The number of functions contained in an AMR with an entropy $e$ and $n$ arguments, including the ones registered explicitly and the ones that are due to the side-effects of the $\lambda$-operation, is $2^{en}$, as specified in Section 1. For instance, Fig 8 (e) shows that $e = 4.5$ for an AMR of size $64 \times 64$ filled up with 772 functions, i.e., an average of 22.62 cells set on per column. The number of functions is obtained considering that the AMR includes 64% of the *RemCorpus*, which is the 33% of the full EMNIST-36 corpus, although the corpus is not balanced and 772 is the average number of functions in the 36 AMRs. Hence, the AMR includes $22.62^{64}$ functions representing actual and potential calligraphic symbols, with satisfactory precision and recall. As can be seen the number of functions input explicitly is very small in relation to the number of potential objects stored in the AMR, proving a very large space for the construction of novel objects and "imagination".

## 5.10 Quantitative comparisons

The present discussion shows that there are several qualitative dimensions that should be considered for comparing the present proposal with associative memories developed within the neural networks paradigm. One preliminary consideration is that artificial neural networks are often used to model functions that differ from memory, such as perception and action. However, these are often labeled as "memories", for instance in neuroscience studies [30]. However, for something to be a memory at the functional level [31] the objects registered should be stored in a format that allows that such objects can be retrieved or remembered later. In the present approach we use neural networks to implement perception and action, but these are different from the memory itself, both structurally and functionally. Of course, the performance of the memory system depends on the performance of the networks implementing the analysis and synthesis modules, but comparisons to asses the present proposal must be with systems that implement associative memories at the functional level. In particular comparing the functional with the algorithmic or implementation level in Mars's sense [31] is a category mistake. Hence, neural networks as well as other kinds of machine learning mechanisms implementing classifiers, filters, controllers, etc., should not be considered memories. An example is the use of celullar automata to classify the digits of the MNIST corpus [32]: this system can tell the class of a numeral but cannot use its full or incomplete image as a cue to retrieving the digit. Conversely, a memory system can be used as a classifier, as Krotov and

Hopfield's dense associative memories [29] for classifying the MNIST corpus too, but in such model, the memory system is used at an algorithmic level to support classification at the functional level, reversing somehow Marr's hierarchy of system levels.

Comparisons also face the problem that most proposals are tested with a corpus suited for their particular goals; so corpora designed for a particular model or application cannot be used directly in other settings. In particular we considered associative memories that use MNIST or EMNIST, such as the spiking Neural Networks developed by Hu et al. [26]. However, the experiments are limited to a very small set of cases, and cannot be compared directly with ours. We also reviewed the survey of systems using MNIST and EMNIST for handwritten character recognition [33], but associative memories are not even mentioned, and we were unable to find a single associative memory using the EMNIST corpus for direct comparisons.

The most similar study that we found was Krotov and Hopfied's dense associative memories [29] for classifying the MNIST corpus. We adapted such approach using the code available at https://github.com/DimaKrotov/Dense_Associative_Memory to make a quantitative comparison with EAM, but testing with the EMNIST-36 corpus instead. Dense associative memories extend Hopfield's original model using a new energy function with a much larger number of local minima and a much larger storing capacity. For their experiments the corpus was partitioned in the training and testing corpora, which are mutually exclusive, as is done in standard classifiers. The training corpus was used to train the memory and the test corpus to asses the classification performance. For our experiment we used the 112,800 and 18,800 instances images of the training and test corpora of EMNIST-36, respectively; the corresponding sets of MNIST consists of 60,000 and 10,000 instances, respectively. We computed the precision of the training and test corpora using their methodology, rendering 0.7957 and 0.7813, respectively. These results can be contrasted with our experiment 3.2 in Fig 8 (f) using the totality of remembered corpus, where precision is 0.8868 and recall is 0.8361, which improves over the figures obtained using Krotov and Hopfield's code. However, it should also be considered that the classification rate for this task using standard classifiers may be much higher; in particular our classifier in Fig 4 renders an accuracy of 0.9, and suggests that an associative memory system is not the best tool for performing classification.

This comparison should also be placed in the larger perspective of the nature and assumptions of the two kind of memory systems. Hopfiels' model is the paradigmatic model of auto-associative memories, and the memory is reproductive. Consequently, in the classification exercise, all of the cues in the test corpus fall into the most proximate local minima but never select the right object, which is not in the training set, hence in the memory, and should count as false positives. Conversely, as Hopefiel's memory always selects the most proximate object, there are no rejections. EAM, in contrast, assumes that cues are never contained in the memory exactly, unless the entropy is very low, the precision very high, and the recall very low, which is an exceptional situation, and even then the retrieved object is a construction that may differ slightly from the cue. This highlights that Hopfield's memory is a pattern matching machine at the computational or functional level in Marr's sense [31] that is used pragmatically as a memory, while EAM aims to model natural memory at such system level, in a manner coherent with the algorithmic and implementational levels.

Comparison with other associative memories within the neural networks paradigm, such as Associative Long-Short Term Memories [34], could be made –although neither their corpus nor their code are available– but in any case, considerations similar to the ones made above for dense associative memories, would have to be taken into account. Most generally, EAM uses a declarative format holding a distributed representation that is manipulated symbolically, and differs from the conception of associative memory as a dynamical system with an energy function, and a direct comparison between these two approaches constitutes a category mistake.

## 6 Summary and discussion

The present memory system is associative because the AMRs are accessed through their contents, which are codified as discrete functions represented in a tabular format. The concrete representations of manuscript symbols are placed on modality specific input and output buffers, and these are translated to their abstract representations, which are input to the AMRs through the encoder. The output of the AMR is an abstract representation of the same kind that is fed into the decoder to generate the corresponding concrete representation.

EAM holds distributed representations because the individual instances of the stored objects are represented as functions that are overlapped within the corresponding AMR. Hence the cells of the register's table can contribute to the representation of more than one object –all the objects whose representation share the same value for the same argument– and the representation of an object –a function– shares memory cells with the representations of other objects, and the relation between memory units and their contents is *many-to-many* in opposition of local representations in which such relation is *one-to-one* [2].

The memory register operation is conceptualized as an abstraction between the representation of the object to be stored and the content of the memory, and is implemented with the standard inclusive logical disjunction between the arguments and values of the object to be stored and its corresponding arguments and values in the memory, which can be construed as the micro-features of the representations.

The memory recognition operation is conceptualized as the inclusion relation of the object to be recognized in relation to the content of the memory, and is implemented through the logical material implication between the corresponding micro-features of the object to be recognized and the content of the memory. Failure in the recognition test implements a strong negation directly, without search.

The memory retrieval operation is conceptualized as a constructive operation that renders a novel object always. The retrieval operation is conditioned by the recognition test, and the rendered object is built by selecting randomly the values associated to all the arguments of the function representing the retrieved object, using a distribution centered on the memory cue.

The objects of computing in the three memory operations are the micro-features constituting the representations. The computations can be performed in parallel in a natural way taking very few computing steps, if the appropriate hardware is made available.

The memory operations conform to the main properties of human memory that emerged from the paradigmatic studies carried on by Bartlett [8]; and differ fundamentally from the corresponding operations in memories developed within the neural networks paradigm, as was intensively discussed in Section 5.

AMRs hold relations that have a certain amount of indeterminacy. A function is a fully determined relation and its entropy is zero. The entropy is related to the number of objects stored in the memory; when there are few, the entropy is very low, and successful retrieval operations are very precise but recall is very low; conversely, when the number of stored objects is high, so is the entropy, and recall may be very high, but precision decreases significantly. The entropy depends on the interactions between such functions, i.e., the number of arguments that share the same value for a number of functions: the greater the interactions the lower of the entropy. More generally, the operational capacity of AMRs conforms to an entropy trade-off, to the effect that the objects recovered through the memory retrieval operation in relation to the memory cue are "photographic reproductions", "appropriate reconstructions", "imaged objects" and noise, according to the increase of the entropy, from very low to very high, respectively. The trade-off between precision and recall is satisfactory for moderate entropy values.

The purpose of experiments 1 and 2 was to identify the size of operational AMRs for the application domain. Our previous work with digits showed that AMRs with 64 features –the cardinality of the domain of the stored functions– offer a good compromise between performance and memory size and, consequently, cost [1]. Further experiments showed that this choice is appropriate for the present case study too. Figs 7 and 8 show the performance of the memory for different sizes or discrete levels of the functions' codomains. This choice is also essential for the system performance and its cost. The tests were performed with ten values of the parameter $m$, where the number of rows of the AMRs is $2^m$. Figs 7 and 8 from (a) to (d) in experiments 1 and 2 show that there is a good compromise between precision and recall at $m = 6$ and $m = 7$ with entropy from 4.6 to 5.4 in both experiments.

Then, we used the test corpus to measure the performance of the system when the operational registers have different amounts of information. Figs 7 and 8, (e) and (f), show the performance of the AMRs with size $64 \times 64$ for the corpora *EMNIST−47* and *EMNIST−36*, respectively; and (g) and (h) the corresponding performance of the AMRs of size $64 \times 128$. As can be seen, a satisfactory performance is achieved with a significant amount of memorized information.

The construction of practical applications requires improving the performance of the system as a whole and reduce the memory size for an arbitrary dataset. In order to address such question, we plan we plan to reinforce the cells of the AMRs whenever they are used in the memory register operation, such that columns become probability distributions shaped by the empirical data, with their associate Shannon's entropy. Such learning mechanism should improve the performance and reduce the size of operational AMRs. We leave such investigation for further work.

AMRs can hold the representation of more than one class, with the only penalty of an increase of the entropy, as shown in our previous work [1]. Here we capitalize such property to represent capital and lower case letters in the same AMRs, reducing the number of registers from 47 to 36. As can be seen in the figures, the reduction in the number of classes compensated the entropy increase of the abstracted classes, and the performance of the system using the EMNIST-47 and the EMNIST-36 is practically the same. Also, the precision and recall are very similar when the AMRs are filled up with a substantial amount of corpus, but considering that the memory cost is twice as much for registers of larger size i.e., $64 \times 128$, we selected the EMNIST-36 corpus and alphabet, with AMRs of size $64 \times 64$ for the memory retrieval experiments.

Although the focus of the experiments presented in this paper is the auto-associative aspect of EAM, the corpus and the experiments do show hetero-associations between capital and their corresponding lower-case letter symbols. However, EAM cannot be compared directly with systems in which the associations are stated explicitly beforehand, such as BAM and morphological models, because in the present proposal all inputs are independent, and the associations are established dynamical as side-effects of the λ-operation, the diagrammatic representation of functions and relations, and the stochastic aspect of the β-operation. For the moment, the hetero-associative aspect of EAM is left for further work.

Experiment 3 confirmed our previous memory retrieval experiments using memory cues that are recovered at very low entropy, moderate entropy but not at lower entropy, and recovered only at high entropy or not recovered at all, as shown in Figs 9–11, respectively, that confirms the entropy trade-all: the reproductions of cues recovered at very low entropy have high quality, but there are very few objects in the memory; cues recovered at moderate entropy can be reconstructed flexibly and with a reasonable quality; and cues that are only recovered at high entropy or not recovered at all correspond to objects of bad quality or that are not included in the memory at all.

However, when the output of the encoder is fed into the decoder directly, an object is rendered always, independently of its quality. If the cue is poor or very poor, the shape rendered by the decoder is due to the prototype patterns codified implicitly in the networks implementing the analysis and synthesis modules.

Experiment 4 addresses recovering objects with incomplete information. In this case, with severely occluded objects. This task has been an important motivation for auto-associative memories. In most realistic situations, where objects are seen from different orientations and distances, and there is usually noise, such as occlusions, poor lighting conditions, or impaired vision, the cue is incomplete, and it has to be approximated to the object recovered in the memory. If the cue is too poor the retrieved object may be the wrong one, and the agent would have no means to realize such a failure, unless the object is subjected to further contextual interpretation. Furthermore, if the cue is not included in the memory it should be rejected directly, as in EAM, in opposition to models that adopt the Closed-World Assumption, and produced the most similar object, which in this case is a false positive.

Incomplete information of the cue is reflected in the values of a small number of features of the function representing the object, which are the cause of the rejection by the recognition test [1]. Hence, our strategy to recover objects with incomplete cues consists simply of relaxing the recognition test. Figs 12 and 13 show the performance of the retrieval operation with very severely occluded objects in two settings –covering the bottom-half of the figure and placing the figure behind wide bars. These conditions are very hard to interpret even for people, and constitute a very strong test for memory systems in general.

The results show another aspect of the entropy trade-off. Most objects severely occluded are likely to be rejected at low entropy levels, as shown in the tables (a) in Figs 12 and 13. However, a small relaxation of 1, 2 and 3 features increases the recall proportionally, and objects may be recovered at lower levels of entropy, as shown in Figs 12 and 13, (b), (c) and (d), reducing the gap of blanks at low entropy levels. As before, the price to pay for the increase of the recall is a decrease of the precision, and the objects recovered may be of the wrong class. Nevertheless, having some candidate interpretations that can be processed further in relation to a context is better than having none.

The Entropic Associative Memory has a variety of potential applications in which a large amount of data needs to be accessed by content. For instance, the present study and experiments can be applied to the construction of handwriting optical character recognition, which is useful for scanning manuscript texts. The memory system can be applied to computer vision and speech processing, and some preliminary experiments are already on their way. More generally, an effective associative memory should be a central functional module of the cognitive architecture of computational agents that need to interact in real time with the world.

The present research introduces a novel application to autoencoders developed within the deep-neural networks paradigm. Such systems were introduced to reduce the number of dimensions of large features spaces, but not to produce abstract representations of concrete objects and vice versa, as is performed by the analysis and synthesis modules of the EAM architecture, respectively. The construction of this architecture requires making such modules independent, so the output of the encoder and the input to the decoder, which are the objects of the memory operations, are independent.

Finally, the present experiments show that the entropic associative memory has a satisfactory performance for storing, recognizing and retrieving manuscript letters and digits; and can potentially be used in practical applications. The present theory and experiments also show a novel conception of memory that differs in several respects from the neural networks paradigm. In particular, it uses a more transparent notion of distributed representation; makes an explicit use of the entropy; and shows that memory conforms to the entropy trade-off. The

memory operations use the logical disjunction, the material implication, and the strong negation directly, which operate on the micro-features of representations, and can be computed in parallel in a very reduced number of steps if the appropriate hardware is provided. This suggests the conjecture that such logical functions have their roots on the basic operations of associative memory. The random element suggests that imagination, creativity and free will have their roots, at least in part, in associative memory too. The memory is constructive as opposed to reproductive, and resembles better the properties of natural memory within constructivist approaches to knowledge and learning.

## Author Contributions

**Conceptualization:** Luis A. Pineda.

**Data curation:** Noé Hernández, Victor D. Cruz.

**Formal analysis:** Rafael Morales, Luis A. Pineda.

**Funding acquisition:** Luis A. Pineda.

**Investigation:** Rafael Morales, Luis A. Pineda.

**Methodology:** Rafael Morales, Luis A. Pineda.

**Project administration:** Luis A. Pineda.

**Resources:** Luis A. Pineda.

**Software:** Rafael Morales, Noé Hernández, Ricardo Cruz, Victor D. Cruz.

**Supervision:** Rafael Morales, Luis A. Pineda.

**Validation:** Rafael Morales, Noé Hernández, Ricardo Cruz, Victor D. Cruz, Luis A. Pineda.

**Visualization:** Rafael Morales, Noé Hernández, Ricardo Cruz, Victor D. Cruz.

**Writing – original draft:** Luis A. Pineda.

**Writing – review & editing:** Rafael Morales, Noé Hernández, Ricardo Cruz, Victor D. Cruz, Luis A. Pineda.

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
