## [Decision Letter · Decision Letter 0]

20 Jun 2022

PONE-D-22-03590Entropic Associative Memory for Manuscript SymbolsPLOS ONE

Dear Dr. Pineda,

Thank you for submitting your manuscript to PLOS ONE. After careful consideration, we feel that it has merit but does not fully meet PLOS ONE’s publication criteria as it currently stands. Therefore, we invite you to submit a revised version of the manuscript that addresses the points raised during the review process.

ACADEMIC EDITOR:

We look forward to receiving your revised manuscript.

Kind regards,

Talib Al-Ameri, Ph.D

Academic Editor

PLOS ONE

Journal Requirements:

When submitting your revision, we need you to address these additional requirement.

“NO”

“Luis A. Pineda acknowledges the partial support of grant PAPIIT-UNAM IN112819, Mexico “

Universidad Nacional Autónoma de México

PAPIIT-N112819

Dr. Luis A. Pineda

Reviewers' comments:

Reviewer's Responses to Questions

**Comments to the Author**

1. Is the manuscript technically sound, and do the data support the conclusions?

Reviewer #1: Yes

Reviewer #2: Yes

2. Has the statistical analysis been performed appropriately and rigorously? 

Reviewer #1: Yes

Reviewer #2: Yes

3. Have the authors made all data underlying the findings in their manuscript fully available?

Reviewer #1: Yes

Reviewer #2: Yes

4. Is the manuscript presented in an intelligible fashion and written in standard English?

Reviewer #1: Yes

Reviewer #2: Yes

5. Review Comments to the Author

Reviewer #1: The authors investigated the entropic associative memory for manuscript symbols. This topic is interesting and shall be worth studying. This paper is well written. Please sFor the comment, please see the attached file.

Reviewer #2: This paper mainly studies that manuscript symbols can be stored, recognized and retrieved from an entropic digital memory that is associative and distributed but yet declarative; memory retrieval is a constructive operation, memory cues to objects not contained in the memory are rejected directly without search, and memory operations can be performed through parallel computations. Manuscript symbols, both letters and numerals, are represented in Associative Memory Registers that have an associated entropy. The memory recognition operation obeys an entropy trade-off between precision and recall, and the entropy level impacts on the quality of the objects recovered through the memory retrieval operation. The technical content of the manuscript seems to be interesting for future application, there are some problems (see the following list) that needs to improve the present work.

1. On page 4, the amount of data allocated by Training Corpus, Remembered Corpus and Test Corpus is 57%, 33% and 10%，respectively. Please explain in detail why such a proportion is allocated and the advantages of this allocation.

2. In Figure 7, when the totality of the remembered corpus is used, the recall is lower and the graphs do not intersect. For the largest registers, the cost of memory resources is twice as high. How to solve these problems in practice.

3. The language needs to be polished and some typos should be corrected. Please check the language and improve the presentation.

4. Please explain the purpose and difference of Experiment 1 and Experiment 2, as well as the similarities and differences of simulation results.

5. On page 10, the sentence “Such is the case of morphological hetero-associative memories [19, 20, 22] and related work [25] which are also bidirectional.” The author can refer to several works of associative memory as bedding, such as Memristor-based neural network circuit of emotion congruent memory with mental fatigue and emotion inhibition. Memristor-based neural network circuit of pavlov associative memory with dual mode switching. Memristor-Based Neural Network Circuit of Full-Function Pavlov Associative Memory with Time Delay and Variable Learning Rate.

6. How to set at about 4% of the remembered corpus or at an entry level between 3 and 4 in the second condition of memory retrieval operation, and what are the advantages of such setting.

6. PLOS authors have the option to publish the peer review history of their article (what does this mean?). If published, this will include your full peer review and any attached files.

Reviewer #1: No

Reviewer #2: No

---

## [Author Response · Author response to Decision Letter 0]

28 Jun 2022

June 26th, 2022

Talib Al-Ameri, Ph.D

Academic Editor

PLOS-ONE,

Dear Prof. Al-Ameri,

We have uploaded the revised version of the paper Entropic Associative Memory for Manuscript Symbols, PONE-D-22-03590. We have attended to all the comments made by the two reviewers and include the relevant suggested references. We also removed the acknowledgements section and include the funding information as required.

The answers to reviewer 1 are:

1. The motivation of this work shall be improved.

Response: The motivation of the work is to add evidence on the viability of the EAM model and its associated theory for developing computational models of natural memories as well as practical applications. This is emphasized in lines 3-6 of the 1st paragraph of Section 1 , which now reads: “Our motivation is to add evidence on the viability of the EAM model and its associated theory for developing computational models of natural memory and to show its feasibility for constructing practical applications.”

2. What is the novelty of this article? What is the technical difficulty on methods and analysis aspect? Please give some explanations.

Response: This paper introduces explicitly the capacity of an Associative Memory Register (AMR) at a given state as a function of the entropy e and the number of columns n, which is 2en. The impact of this measure is discussed explicitly in Section 5.9. We also show a case study in which instances of manuscripts symbols of the EMINIST Corpus are stored and retrieved. The approach to the problem is novel, and an extensive comparison with related work is presented in Section 5.10. An important characteristic of the model is its simplicity, as the memory has a declarative format. To appreciate this, we include the formal definition of the memory operations at the end of Section 1 in p.3 and p.4 –instead of just making a reference to the original EAM publication.

3. The practical application of entropic associative memory for manuscript symbols shall be presented in Conclusion.

Response: A new paragraph has been added in Section 6, p. 23 as follows: “The Entropic Associative Memory has a variety of potential applications in which a large amount of data needs to be accessed by content. For instance, the present study and experiments can be applied to the construction of handwriting optical character recognition, which is useful for scanning manuscript texts. The memory system can be applied to computer vision and speech processing, and some preliminary experiments are already on their way. More generally, an effective associative memory should be a central functional module of the cognitive architecture of computational agents that need to interact in real time with the world.”

4. Compared with the previous publications, what is the advantage of this paper?

Response: The experiments presented in this paper deal with a much larger number of classes than the previous one: 47 and 36 versus 10 and 5, respectively, with satisfactory results. In addition, a detailed comparison in 10 dimensions with previous work in associative memories developed within the framework of Artificial Neural Networks (ANNs) is presented in Section 5. These include (1) EAM’s the declarative format versus the procedural or sub-symbolic format of ANNs; (2) the productive or generalization property of EAM that ANNs’ memories lack; (3) the constructive retrieval of EAM versus the photographic retrieval of ANNs approaches; (4) the integrated auto and hetero associativity of EAM; (5) the property of direct rejection of EAM that ANNs approaches lack; (6) the natural parallelism of EAM due to the direct cell to cell and column operations, which does not involve matrix operations; (7) EAM is not a dynamic system and does not use an energy function; (8) the explicit use of the entropy by EAM that ANNs approaches lack; (9) the very large storing capacity of EAM; and (10) the quantitative comparisons of EAM with ANNs; in particular, with associative memory systems using similar corpus, such as dense associative memories (Krotov and Hopfield, Ref. 29).

5. The format of the references shall be united

Response: All references have been reviewed and included using the Latex BibTex format, using PLOS ONE templates.

6. The following related publications on associative memory shall be added:

Response: Our paper is concerned with the performance of a novel model of associative memory that uses a declarative format in which memory register is an abstraction, cues not contained in the memory are rejected directly without search, and memory retrieval is a constructive operation. We assessed our model empirically in terms of the precision, the recall and the accuracy of the memory recognition and the memory retrieval operations in relation to the memory size, the amount remembered data and the entropy, and the tolerance to error; and we offered a comparison with related work addressing similar problems. We could not find any of these concerns in the recommended papers and are unable to appreciate why such references shall be included, and we have not done so.

The answers to reviewer 2 are:

1. On page 4, the amount of data allocated by Training Corpus, Remembered Corpus and Test Corpus is 57%, 30% and 10%, respectively. Please explain in detail why such a proportion is allocated and the advantages of this allocation.

Response: A new paragraph was added in Section 3, p.7 as follows: “The data allocated to each partition reflects a trade-off between learning and test data, so 90% of the corpus is used for the former and 10% for the latter, according to standard machine learning practices. The balance between training and remembered data considers that a large enough amount of corpus is needed for training the deep neural networks modeling perception and action, but enough data is needed to test the AMRs with different remembering conditions and entropy levels. We also made preliminary experiments with small variations to these amounts, and the present choice constitutes a satisfactory compromise.”

2. In Figure 7, when the totality of the remembered corpus is used, the recall is lower and the graphs do not intersect. For the largest registers, the cost of memory resources is twice as high. How to solve these problems in practice.

Response: The fact that the precision and the recall do not intersect with a given amount of data is not necessarily a problem in itself. Figure 7 shows that grids of 64 and 128 rows perform better for the EMNIST dataset than smaller and larger grids, and that the best performance is achieved using the totality of the remembered data. Hence these sizes are considered for functional AMRs and are tested with different amounts of corpus and entropy levels. The graphs show that the best performance for this dataset is achieved using the totality of the remembered corpus: the results also suggest that registering more information will not impact on the performance significantly. As the performance is very similar in both settings the smallest AMRs size is chosen due to its reduced cost. A related but different question is how to improve the performance of the system as a whole and reduce the memory size for an arbitrary dataset. For this we plan to reinforce the cells of the AMRs whenever they are used in the memory register mechanism, such that columns become probability distributions shaped by the empirical data, with their associate Shannon's entropy. Such learning mechanism should improve the performance and reduce the size of operational AMRs. We leave such investigation for further work.

A new paragraph addressing this latter question has been added in Section 6, p. 22, as follows: “The construction of practical applications requires improving the performance of the system as a whole and reduce the memory size for an arbitrary dataset. In order to address such question, we plan we plan to reinforce the cells of the AMRs whenever they are used in the memory register operation, such that columns become probability distributions shaped by the empirical data, with their associate Shannon's entropy. Such learning mechanism should improve the performance and reduce the size of operational AMRs. We leave such investigation for further work.”

3. The language needs to be polished and some typos should be corrected. Please check the language and improve the presentation.

Response: The language has been revised and the whole of the paper has been carefully proof-read.

4. Please explain the purpose and difference of Experiment 1 and Experiment 2, as well as the similarities and differences of simulation results.

Response: The first line of paragraph of section 3.2, p.10, has been elaborated to clarify these questions as follows:

“This experiment shows that an AMR can hold the distributed representation of objects of different classes, such as the representations of different digits –e.g., 0 and 1– with similar levels of precision and recall but a small increment of the entropy, as was shown in our previous work. In this case, we use EMNIST-36 instead of EMNIST-47 such that eleven different shaped capital and lower case letters are collapsed in the same class.”

5. On page 10, the sentence “Such is the case of morphological hetero-associative memories [19, 20, 22] and related work [25] which are also bidirectional.” The author can refer to several works of associative memory as bedding, such as Memristor-based neural network circuit of emotion congruent memory with mental fatigue and emotion inhibition. Memristor-based neural network circuit of Pavlov associative memory with dual mode switching. Memristor-Based Neural Network Circuit of Full-Function Pavlov Associative Memory with Time Delay and Variable Learning Rate.

Response: Two of the suggested references have been included in Section 5, p. 16 (Refs. 27 and 28).

6. How to set at about 4% of the remembered corpus or at an entry level between 3 and 4 in the second condition of memory retrieval operation, and what are the advantages of such setting.

Response: The amount of corpus fed into the AMR is set up in the experimental design as desired. Figures 7 and 8 show that for AMRs filled in with 3% to 4% of the corpus the precision is close to 1 but the recall is almost zero, both for grids of 64 x 64 and 64 x 128, for both the individual AMRs and the system as a whole. The reason is that the amount of information within the AMR is very low, so if a cue retrieves an object, it is almost certain that it is of the right class, but most AMR do not respond to the cue, and most true positives are missed out.

We hope that the new version of the paper is fully satisfactory.

Yours sincerely

Dr. Luis A. Pineda

Titular Research Professor

Institute for Applied Mathematics and Systems

Universidad Nacional Autónoma de México

---

## [Decision Letter · Decision Letter 1]

19 Jul 2022

Entropic Associative Memory for Manuscript Symbols

PONE-D-22-03590R1

Dear Dr. Pineda,

We’re pleased to inform you that your manuscript has been judged scientifically suitable for publication and will be formally accepted for publication once it meets all outstanding technical requirements.

Kind regards,

Talib Al-Ameri, Ph.D

Academic Editor

PLOS ONE

Reviewers' comments:

Reviewer's Responses to Questions

**Comments to the Author**

1. If the authors have adequately addressed your comments raised in a previous round of review and you feel that this manuscript is now acceptable for publication, you may indicate that here to bypass the “Comments to the Author” section, enter your conflict of interest statement in the “Confidential to Editor” section, and submit your "Accept" recommendation.

Reviewer #1: All comments have been addressed

Reviewer #2: All comments have been addressed

2. Is the manuscript technically sound, and do the data support the conclusions?

Reviewer #1: Yes

Reviewer #2: Yes

3. Has the statistical analysis been performed appropriately and rigorously? 

Reviewer #1: Yes

Reviewer #2: Yes

4. Have the authors made all data underlying the findings in their manuscript fully available?

Reviewer #1: Yes

Reviewer #2: Yes

5. Is the manuscript presented in an intelligible fashion and written in standard English?

Reviewer #1: Yes

Reviewer #2: Yes

6. Review Comments to the Author

Reviewer #1: All the questions have been revised and i am satisfied with this version. So this paper can be accepted.

Reviewer #2: Manuscript symbols, both letters and numerals, are represented in Associative Memory Registers that have an associated entropy. It is OK now after this revision.

7. PLOS authors have the option to publish the peer review history of their article (what does this mean?). If published, this will include your full peer review and any attached files.

Reviewer #1: No

Reviewer #2: No

---

## [Editor Report · Acceptance letter]

26 Jul 2022

PONE-D-22-03590R1 

Entropic Associative Memory for Manuscript Symbols 

Dear Dr. Pineda:

I'm pleased to inform you that your manuscript has been deemed suitable for publication in PLOS ONE. Congratulations! Your manuscript is now with our production department. 

Kind regards, 

on behalf of

Dr. Talib Al-Ameri 

Academic Editor

PLOS ONE